# Influence of Perceived Stress and Stress Coping Adequacy on Multiple Health-Related Lifestyle Behaviors

**DOI:** 10.3390/ijerph19010284

**Published:** 2021-12-28

**Authors:** Nobutaka Hirooka, Takeru Kusano, Shunsuke Kinoshita, Hidetomo Nakamoto

**Affiliations:** Department of General Internal Medicine, Saitama Medical University, Morohongo 38, Moroyama-machi, Saitama 350-0495, Iruma-gun, Japan; t_kusano@saitama-med.ac.jp (T.K.); kinoppi@saitama-med.ac.jp (S.K.); nakamo_h@saitama-med.ac.jp (H.N.)

**Keywords:** stress, stress coping adequacy, lifestyle behaviors, national health promotion

## Abstract

Stress is a primary target of national health promotion efforts such as Healthy Japan in the 21st century (HJ21). However, little is known about how the combination of perceived stress and coping adequacy influence health-related lifestyle behaviors in line with national health promotion. This study assessed the association between combined perceived stress and coping adequacy and multiple health-related lifestyle behaviors in HJ21 practices. This cross-sectional survey that included specialists in health management comprehensively assessed multiple health-related lifestyle behaviors in accordance with HJ21. Total health-related lifestyle behavior scores were calculated and perceived stress and coping adequacy were recorded and categorized into four groups with group 1 to 4 being high to none, and highly adequate to not at all, respectively. The average total lifestyle behavior scores (standard deviation [SD]) were 35.1 (3.5), 33.7 (3.6), 31.8 (3.8), and 30.5 (4.9) for groups 1 to 4 of coping adequacy (*p* < 0.001). Further, individuals who had higher stress coping adequacy had better multiple health-related lifestyle behaviors after adjusting for demographic factors and perceived stress in the linear trend among the groups. Stress coping skills might be an essential target for stress reduction, ultimately leading to health promotion for disease prevention and longevity.

## 1. Introduction

Lifestyle behaviors determine a wide spectrum of diseases. Studies show that noncommunicable diseases (NCDs)—including ischemic heart disease, stroke, cancer, diabetes, hypertension, and dyslipidemia—are all associated with poor lifestyle behaviors [1,2,3,4]. Many studies report successful interventions involving lifestyle behaviors to decrease the incidence of NCDs and reduce countries’ societal and economic burdens [5,6,7,8]. In rapidly aging societies such as Japan, the burdens of such diseases are significant and have been growing [9,10]. Thus, promoting healthier lifestyle behaviors and thereby controlling NCDs to stabilize the national health budget is of critical interest in Japan and other countries.

The relationship between stress and lifestyle behaviors can be bidirectional. Stressful situations worsen health-related lifestyle behaviors, such as maladaptive smoking or excessive alcohol intake [11,12,13]. Conversely, some studies indicate that some lifestyle behaviors reduce stress [14,15]. Further, the successful adoption of lifestyle behaviors improves stressful situations, some of which are enabled by coping mechanisms related to stress [14,16,17,18]. Therefore, successful coping to reduce stress is considered an important path to healthier lifestyle behaviors. 

Coping is defined as the cognitive and behavioral efforts of an individual to manage the internal and external demands encountered during a specific stressful situation [19]. Coping is an important factor that influences health-related lifestyle behaviors. Existing studies show a positive association between coping and lifestyle behaviors, such as exercise, physical activity, and substance use [16,20,21,22,23]. Perceived stress has also been associated with coping [24,25]. Some studies indicate that the relationship between perceived stress and physical health is mediated by coping [19]. 

However, this has not been thoroughly verified in the context of health promotion through lifestyle behaviors. Additionally, most previous investigations have focused on an isolated behavior, rather than multiple lifestyle behaviors, which are the real-world targets of health promotion. This study, therefore, aimed to investigate the association of combined perceived stress and coping adequacy with multiple health-related lifestyle behaviors among Japanese health management specialists, who acquired knowledge of healthy lifestyle behaviors and potentially gained coping adequacy skills in line with national health promotion practices. Exploring the mechanistic link between perceived stress, coping adequacy, and multiple health-related lifestyle behaviors in the context of national health promotion provides important insights to help improve public health practice, and thereby national health. 

## 2. Methods

### 2.1. Study Design

This cross-sectional study involved nationally certified specialists in health management. The study population was obtained from the register of the Japanese Association of Preventive Medicine for Adult Disease (JAPA) [26]. We administered a survey on multiple health-related lifestyle behaviors based on the Healthy Japan in the 21st century (HJ21) goals for December 2018 to March 2019. The National Health and Nutrition Survey (NHNS) is the oldest national health examination survey currently conducted worldwide and serves as a primary national database of risk factors for noncommunicable diseases in Japan [27]. The HJ21 used the results of NHNS data and also evidence from other studies to set the indices as goals for lifestyle behaviors to promote health. The survey also included questions concerning demographic data, self-perceived stress, and stress coping adequacy. This study was approved by the ethical committee of Saitama Medical University (ID 896, 2018). Informed consent was obtained from all study participants. 

### 2.2. Study Participants

Study participants are specialists in health management who have the relevant qualifications and who have been certified by the JPCA and supported by the Ministry of Education, Culture, Sports, Science, and Technology. The certification verifies that the individual possesses a high level of health-related knowledge and skills and can make complex health-related decisions. Their proficiency is obtained through the learning process of becoming specialists in health promotion and disease prevention, and through activities in line with HJ21. Certified specialists are expected to engage with the communities and societies in which they live and to promote health using the knowledge and skills they acquire during their initial education, certification process, and continuing education [28]. We included all professionals who actively maintained their knowledge and skills provided by JAPA. There were no specific exclusion criteria. A sample was obtained in a non-probability sampling manner. Among these individuals (*N* = 9149), our final sample comprised 4820 certified professionals who agreed to participate in the study.

### 2.3. Variables and Measurements

The variables measured in this study included demographic data, multiple health-related lifestyle behaviors, perceived stress, and stress coping adequacy. Multiple health-related lifestyle behaviors included diet and nutrition, exercise and physical activity, sleep, rest, smoking, and alcohol intake. These were assessed using a self-administered questionnaire in the same format as the NHNS. The survey comprised ten health-related lifestyle questions, of which five (“Intention to maintain ideal weight,” “Exercise,” “Excessive alcohol intake,” “Manage lifestyle to prevent disease,” and “Smoking”) were dichotomous. For these items, a score of “1” was assigned for an unhealthy lifestyle and a score of “4” was assigned for a healthy lifestyle. The remaining five health-related habits (“Reading nutritional information labels,” “Maintaining a balanced diet in daily life,” “Intention for exercise,” “Rest,” and “Sleep”) were to be answered on a four-point Likert-type scale, from “4” (most favorable) to “1” (least favorable). Finally, we added together the values of the answers to these questions as participants’ total health-related lifestyle behavior scores (4 to 44). 

Participants were also asked to answer questions related to perceived stress and stress coping adequacy in four categories. The perceived stress category included group 1 (high), group 2 (moderate), group 3 (low), and group 4 (none). Stress coping adequacy included group 1 (highly adequate), group 2 (adequate), group 3 (inadequate), and group 4 (not at all). Considering demographic data, numeric data were obtained in terms of age, weight, height, body mass index (BMI), and alcohol intake. Meanwhile, data regarding sex, smoking habit, perceived stress, and stress coping were treated as either binary or ordinal. 

Descriptive statistics (means and SDs) were used to describe participants’ characteristics. Numeric data were calculated for each group in terms of perceived stress and coping adequacy. A correlation analysis was then performed to determine the strength of the association between perceived stress and coping adequacy by measuring Kendall’s *tau-b* and Spearman’s *rho*. Next, we performed a regression analysis to investigate the relationship between perceived stress and total lifestyle behavior scores and between coping adequacy and total lifestyle behavior scores. A linear regression analysis between perceived stress and total lifestyle behavior score after adjusting for perceived stress was performed. These allowed us to investigate the mediating effect of coping adequacy on the relationship between perceived stress and multiple health-related lifestyle behaviors [29]. A multiple linear regression analysis was used to determine the moderating effect of coping adequacy on the relationship between perceived stress and multiple health-related lifestyle behaviors including interaction term of perceived stress and stress coping adequacy. All statistical tests were two-tailed. *p*-values below 0.05 were considered as statistically significant. IBM SPSS Statistics (version 26.0. IBM Corporation, Armonk, NY, USA) software was used for analysis. 

## 3. Results

The demographic characteristics and the scores on health-related lifestyle behaviors of the study participants are shown in Table 1. In total, 4820 certified health management specialists were included in the analysis. There were 3190 women (66.2%) and 1630 men (33.8%). The mean (SD) age of all study participants was 55.4 (±12.2) years. The majority of the study participants felt stress (20.5%, 54.0%, and 21.8% for high, moderate, and low, respectively), while 16.4% and 57.3% of them felt their coping adequacy highly adequate and adequate. In terms of excessive alcohol intake and smoking, the prevalence was 5.8% and 6.1%, respectively. More than 80% of the study participants perform healthy lifestyle behavior regarding diet (reading nutritional labels and maintaining a balanced diet) and hold positive intentions towards exercise. The average of total lifestyle behavior scores (SDs) was 35.1 (3.5), 33.7 (3.6), 31.8 (3.8), and 30.5 (4.9) for groups 1 to 4 of stress coping adequacy (*p* < 0.001). Additionally, more than 70% of them felt their rest and sleep were adequate. The average scores (SDs) of the total lifestyle behaviors were 31.8 (0.13), 33.5 (0.07), 34.2 (0.12), and 35.1 (0.25) for groups 1 to 4 of the stress groups (*p* < 0.001), respectively.

In terms of the analysis of mediating effect of stress coping adequacy on the relationship between perceived stress and lifestyle behaviors, first we performed correlation analysis among stress coping adequacy, perceived stress and lifestyle behaviors, and then the association of perceived stress-lifestyle behaviors and stress coping adequacy-lifestyle behaviors. Finally, the contingency table among the groups of perceived stress and coping adequacy is shown in Table 2. There was a statistically significant negative association between the two groups (Kendall’s *tab-b* and Spearman’s *rho* were −0.581 and −0.624, respectively, both *p* < 0.001). The result of the simple regression analysis between perceived stress and total lifestyle behavior scores was statistically significant (standardized *β* = 0.226, *R*^2^ = 0.051, *p* < 0.001). Similarly, there was a statistically significant association between coping adequacy and total health-related lifestyle behavior scores (standardized *β* = −0.312, *R*^2^ = 0.097, *p* < 0.001). Regression analyses showed that there was statistically significant association between perceived stress and lifestyle behaviors after adjusting for stress coping adequacy (standardized *β* = −0.051, *R ^2^*= 0.100, *p* = 0.004). 

The results of analyzing the moderating effect of the stress coping adequacy were as follows. While there were statistically significant associations of both perceived stress and coping adequacy with multiple health-related lifestyle behavior scores as previously illustrated, perceived stress was no longer associated with multiple health-related lifestyle behavior scores after controlling for age, sex, and coping adequacy (standardized *β* = 0.013, *p* = 0.446). In addition, the interaction term of perceived stress and coping adequacy did not explain the variability of multiple health-related lifestyle behaviors to statistical significance (standardized *β* = 0.025, *p* = 0.201).

## 4. Discussion

The results of our study indicate that the relationship between perceived stress and multiple health-related lifestyle behaviors were mediated by stress coping adequacy. To the best of our knowledge, this is the first study to show the mediating effect of coping adequacy on the relationship between perceived stress and multiple health-related lifestyle behaviors, in terms of national health promotion. The standard *β* of the perceived stress on total health-related lifestyle behavior scores was 0.226 (*p* < 0.001), as shown in the results. The direct effect of perceived stress on total health-related lifestyle behavior scores dropped from 0.226 to 0.051 when coping adequacy was added to the model, indicating that 77.4% of the relationship between perceived stress and multiple health-related lifestyle behaviors was mediated by coping adequacy. The analysis did not show a moderation effect of stress coping adequacy on the relationship between perceived stress and multiple health-related lifestyle behaviors. 

Coping is defined as the cognitive and behavioral efforts of an individual to manage the internal and external demands encountered during a specific stressful situation [19]. Multiple studies show an association between stress coping and health-related lifestyle behaviors such as exercise, physical activity, and substance use [20,21,22,23]. Stress itself has been shown to be a critical factor in determining health-related lifestyle behaviors and health outcomes [30,31,32]. Researchers have investigated the links between stress and factors such as diet, exercise, and substance use [33]. However, most studies examined the link between stress and singular health-related lifestyle behaviors, without assessing coping. In this study, we investigated the moderation and mediation effects of coping adequacy on the relationship between stress and multiple health-related lifestyle behaviors in health management specialists. The results showed a direct effect, not a moderating effect on the relationship between stress and multiple health-related lifestyle behaviors, of stress coping adequacy to mediate the relationship between stress and multiple health-related lifestyle behaviors. 

The implications of the results are two-fold. First, they provide important insights into the mechanistic link between stress, coping, and lifestyle behaviors. Our results indicate that the mediating effect of stress coping adequacy on stress and multiple health-related lifestyle behaviors support the transactional theory model, proposed by Lasarus and Folkman, in which coping mediates the adverse effects on health [19]. This suggests that the transactional model can be applied in public health practice. Transactional theory posits that coping mediates the adverse effects of stress on health outcomes [34,35]. This model has been instrumental in stress and coping research across multiple fields and disciplines for the past several decades [36]. Moreover, previous studies report the mediating effects of stress coping. Gibbons et al. report the mediating effect of coping on the relationship between stress and burnout [37]. From an education perspective, several studies indicate the mediating effect of coping on the relationship between stress and both psychological and physical health [20,38]. According to transactional theory, positive outcomes can occur in stressful situations through the mediating roles of cognitive processes by which meaning is ascribed to stimuli [39]. Our results support this positive mediating effect of coping adequacy on multiple health-related lifestyle behaviors. In addition, to show the mediating effect of stress coping adequacy, the moderating effect of coping adequacy was excluded from the study. This suggests that coping with stress directly influences multiple health-related lifestyle behaviors while in the meantime coping reduces stress that is associated with lifestyle behaviors. We therefore posit that interventions to enhance stress coping in health promotion can have a direct effect on the targeted population. While future research is needed to confirm the causal link between stress and multiple health-related lifestyle behaviors mediated by stress coping, the study extended the evidence of the mediation effect on perceived stress and multiple health-related lifestyle behaviors in health promotion practice. 

Considering our second implication, these results are shown in the context of health promotion. Thus, the results provide meaningful insights in the context of national health promotion, in which improving multiple health-related lifestyle behaviors is emphasized as an intervention for the entire population. Considering that stress has been shown as a significant risk factor for developing NCDs, stress should be considered in health promotion practice [33,40,41]. Stress coping associated with better multiple health-related lifestyle behaviors shown in the study suggests a potential reduction of NCDs in the population, as it is well documented that healthy lifestyle behaviors are critical for reducing the occurrence of NCDs such as hypertension, diabetes, stroke, coronary heart disease, and cancer [42,43,44,45]. Stress coping is a major target in health promotion worldwide [46]. The World Health Organization states that the promotion, protection, and restoration of mental health, including stress coping, is regarded as a vital concern for individuals, communities, and societies globally [47]; therefore, the organization supports governments in strengthening and promoting stress management [48]. In Japan, the HJ21 national health promotion initiative was issued by the Japanese government in 2001 to achieve a vibrant society with healthy and spiritually rich lives, enabled by improvements in lifestyle and social environment [49]. The Ministry of Health, Labor, and Welfare subsequently selected relevant targets for improving lifestyle-related diseases [50]. Mental health, including stress and coping, was set as one of the nine targets in the original HJ21 [50]. In this study, we used the questionnaires recommended in the HJ21 to measure multiple health-related lifestyle behaviors, stress, and coping; our results indicate that the direct effect of coping adequacy more than the perceived stress on health-related lifestyle behaviors as shown in the results strengthens the national policy of HJ21 practice throughout the nation. While it is necessary to expand the external validity of the results, our results may provide important insights when planning national health promotion and considering stress coping as a meaningful target.

Despite the clear value of its implications, this study has several limitations that must be noted. First, there might be a sampling bias since the sampling was non-probability method. While more than half of the population who met the inclusion criteria participated in the study and 100 percent of the response, it is hoped that studies with less systematic errors confirm the results in the future. Second, the self-reported nature of the data may have induced reporting bias; therefore, future studies should objectively measure and analyze the data. Third, the design was cross-sectional; therefore, correlational and longitudinal studies are needed to investigate the causal link between stress coping adequacy and multiple health-related lifestyle behaviors. Lastly, the study population was restricted to health-literate individuals, as the study participants were specialists in health management. Future studies should target wider populations to investigate the mechanism of the impact of stress coping adequacy on multiple health-related lifestyle behaviors.

Despite these limitations, the nationwide large sample size of our study allowed meaningful analysis to investigate associations among stress, stress coping adequacy, and multiple health-related lifestyle behaviors. Second, stress and stress coping were the two major psychological factors in the frequently-referenced model of transactional theory. The study results that were considered in the analysis have not been tested in the public health practice. This widens the evidence of the transactional model.

## 5. Conclusions

Stress coping adequacy directly influences individuals to determine their multiple health-related lifestyle behaviors, while stress is negatively associated with these behaviors. Stress coping skills may be a critical target for health promotion, in line with national health promotion for disease prevention and healthy longevity.

## Figures and Tables

**Table 1 ijerph-19-00284-t001:** Participants’ demographic characteristics (*N*; %).

Characteristics	*N*	%
Sex:		
Male	1630	33.8
Female	3190	66.2
Age range (years):		
<29	129	2.7
30–39	372	7.7
40–49	930	19.3
50–59	1541	32.0
60–69	1291	26.8
70–79	489	10.1
>80	68	1.4
Age (Ave years, SD)	55.4	±12.2
Height (Ave cm, SD)	161.3	±8.0
Weight (Ave kg, SD)	57.5	±10.8
BMI (Ave kg/m^2^, SD)	21.9	±3.3
Perceived stress:		
High	985	20.5
Moderate	2601	54.0
Low	1049	21.8
None	179	3.7
Coping adequacy:		
Highly adequate	789	16.4
Adequate	2760	57.3
Inadequate	1063	22.1
Not at all	202	4.2
Intention to keep ideal weight	3969	82.6
Managing Lifestyle for disease prevention	4290	89.2
Excessive alcohol intake	279	5.8
Adequate exercise	3079	63.9
Smoking	292	6.1
Reading nutritional information labels:		
Always	1653	34.3
Often	2310	47.9
Rarely	647	13.4
Very rarely	207	4.3
Maintaining a balanced diet in daily life:		
Always	2545	52.8
Often	1830	38.0
Rarely	384	8.0
Very rarely	59	1.2
Intention for exercise:		
Always	2037	42.3
Sometimes	1991	41.3
In the past	638	13.2
Never	150	3.1
Rest:		
Satisfactory	1029	20.5
Adequate	2609	54.0
Not adequate	1020	21.8
Not satisfactory	160	3.7
Sleep:		
Satisfactory	1027	21.3
Adequate	2765	57.4
Not adequate	979	20.3
Not satisfactory	48	1.0
Total lifestyle behavior score (Ave, SD)	33.4	±3.87

Unless otherwise noted, values are number of participants and percentage. Ave: average; *N*: number of participants; SD: standard deviation.

**Table 2 ijerph-19-00284-t002:** Contingency table between the subgroups of perceived stress and coping adequacy.

		Coping Adequacy *N* (%)			
		Group 1	Group 2	Group 3	Group 4
Perceived stress	Group 1	15 (1.5)	304 (30.9)	478 (45.0)	188 (19.1)
	Group 2	148 (5.7)	1881 (72.3)	559 (21.5)	13 (0.5)
	Group 3	461 (43.9)	564 (53.8)	23 (2.2)	1 (0.1)
	Group 4	165 (92.2)	11 (6.1)	3 (1.7)	-

Groups 1 to 4 in perceived stress indicate high, moderate, low, and none, respectively. Groups 1 to 4 in stress coping adequacy indicate highly adequate, adequate, inadequate, and not at all, respectively.

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
