# Peer review of "Influence of Perceived Stress and Stress Coping Adequacy on Multiple Health-Related Lifestyle Behaviors"

_ijerph, 2021, doi:10.3390/ijerph19010284_

Round 1

Reviewer 1 Report

I am confused by the target population, those “certified by the JPCA and supported by the Ministry of Education, Culture, Sports, Science, and Technology.”  If you are investigating lifestyle behaviors that have been used as coping strategies for lessening stress to promote wellness to the general population, then why are you targeting yoru surveys to such a specific group of individuals who may not have behaviors that align with the general population. Leaning about the behaviors of the general population would help drive this public health practice.

In line 92, what is meant by “these data?”  The finding from this study or from the NHNS?  I am not sure why this is relevant here.

The results should be further explained (Specifically linked 136-154).  I also don’t see mention of the specific lifestyle behaviors other than in Table 1.

This claim cannot be made based on the target population of the study “We therefore posit that interventions to enhance stress coping in health promotion can have a direct effect on the entire targeted population”

Author Response

Point by point reply to the reviewers’ comments

Reviewer 1

  1. I am confused by the target population, those “certified by the JPCA and supported by the Ministry of Education, Culture, Sports, Science, and Technology.”  If you are investigating lifestyle behaviors that have been used as coping strategies for lessening stress to promote wellness to the general population, then why are you targeting yoru surveys to such a specific group of individuals who may not have behaviors that align with the general population. Leaning about the behaviors of the general population would help drive this public health practice.

(Reply) Thank you for the comments about the population of our study. As pointed by the reviewer, the study population is specialists in health management, not general population. Thus, implication based on the result applies to the specialist in health management. Despite the discussed generalizability issue, there are several points that we have considered the population we studied. First, there were several studies reporting that health professionals do not maintain healthy lifestyle behaviors (Refs). Thus, even some may think they perform healthier (or unhealthier), it is worth studied on the health professionals who are under stress. Second, the target population had gone through the education in the area of health promotion and potentially acquired the skill of coping adequacy based on the skills they learned and used during the health promotion activities. Public health promotion targets people like the health specialists we studied would gain knowledge, skills, and capability of performing healthy lifestyle as the outcome. Investigating on the specialists in health management, thus, may help delineating the process of changing and mechanism of effect of stress coping on lifestyle behaviors. This is the logic of studying in the target population; specialists in health management. We modified the Introduction to incorporate this point of choosing the population (Line 43-46, Page 2)

  1. In line 92, what is meant by “these data?”  The finding from this study or from the NHNS?  I am not sure why this is relevant here.

(Reply) Thank you for pointing out the relevance of the statement in the variables and measurements section. Considering the comments which we agree with, the statement has moved from variables and measurements to design section explaining the nature of HJ21. (Line 56-58, Page 2)

  1. The results should be further explained (Specifically linked 136-154).  I also don’t see mention of the specific lifestyle behaviors other than in Table 1.

(Reply) Thank you for pointing out the inadequate illustration of the results about each lifestyle behaviors. While the main independent variable in the study was total score of lifestyle behaviors, summary of each lifestyle shown table 1 is important to be explored. Thus, we added further explanation of each lifestyle behaviors in addition to relation between perceived stress and coping adequacy. (Line 105-110, Page 3; Line 119-129, Page

4; Line 134, Page 5)

  1. This claim cannot be made based on the target population of the study “We therefore posit that interventions to enhance stress coping in health promotion can have a direct effect on the entire targeted population”

(Reply) Considering the study participants as replied previously, the statement was modified in the discussion. (Line 174-175, Page 6)

Reviewer 2 Report

I linked a file with suggestions and comments.

Author Response

Reviewer 2

1. This study, therefore, aimed to investigate the association of combined perceived stress and coping adequacy with multiple health-related lifestyle behaviors among Japanese health management specialists, in line with national health promotion practices. Exploring the mechanistic link between perceived stress, coping adequacy and multiple health-related lifestyle behaviors in the context of national health promotion provides important insights to help improve public health practice, and thereby national health. It seems to me that no mechanistic link is explained in the work.

(Reply) Thank you for pointing out the mechanism among stress, stress adequacy, and lifestyle behaviors. We added the discussion of the mechanistic link in the third paragraph of the discussion which generally illustrate the existing theory to explain the association among perceived stress, coping and lifestyle behaviors. (Line 173-174, Page 6)

2. In this study, we used the questionnaires recommended in the HJ21 to measure multiple health-related lifestyle behaviors, stress, and coping; our results indicate that the direct effect of coping adequacy on health-related lifestyle behaviors strengthens the national policy of HJ21 practice throughout the nation. While it is necessary to expand the external validity of the results, our results may provide important insights when planning national health promotion and considering stress coping as a meaningful target. Despite the clear value of its implications, this study has several limitations that must A multiple linear regression analysis was used to determine the moderating effect of coping adequacy on the relationship between perceived stress and multiple health-related lifestyle behaviors.

Observations:

1) There is selection bias, there is no sampling design. The sample isn’t statistical sample. line 81

(Reply) We further explained additionally sampling method. We did not use random sampling when choosing the study participant from the population and can not eliminate selection bias. While majority of the specialists who meet the entry criteria (4820/9149; 53%) participated in the study, we added discussion of potential selection bias in the limitation. (Line 70, Page 2; Line 199-202, Page 6)

2) Sample size aren’t fixed on the basis of expected statistical error line 82

(Reply) Sample size is not calculated. While non-probability sampling as mentioned above, statistical powers to analyze the associations in the study seem to hold adequate power to reach significance. (Line 70, Page 2)

3) The correlation analysis my be inadeguate as the variables considered doesn’t satisfies the normality condition.

(Reply) Since the perceived stress and coping adequacy were categorical variables, we chose correlational analysts applicable to categorical ones; Kendall’s tau-b and Spearman’s rho

4) “The multipler linear regression analysis was used to determine the moderating effect of coping adequacy on the relationship between perceived stress and multiple health-related lifestyle behaviors “ : please clarify , the moderating effect does it mean related to both the perception of stress and the state of health? What is the causal mechanism? line 120 5) “There was a statistically significant negative association between the two groups (Kendall’s tab-b and Spearman’s rho were -0.581 and -0.624, respectively, both p < 0.001).

The results of the simple regression analysis between perceived stress and total lifestyle behavior scores was statistically significant (standardized β= 0.226, R2 = 0.051, p<00001” The R squared indicates that the regressors explain only 5% of the variability of the perception of stress, it seem to me that the indication is weak.

(Reply) We appreciate the reviewer’s comment that the degree of the variability that perceived stress explained lifestyle behaviors was rather weak since the R square 5% (standard β=0.226). Coping adequacy explained more variability of lifestyle behaviors (9.7%; standard |β| = 0.312). Therefore, when considering lifestyle modification, modification of stress coping skills practically is needed in addition to the stress reduction. These were further commented in the discussion in the fourth paragraph. (Line 193-196, Page 6)

Round 2

Reviewer 2 Report

The answer is attached

thank you
